# The Beneficial Effects of Prenatal Biotin Supplementation in a Rat Model of Intrauterine Caloric Restriction to Prevent Cardiometabolic Risk in Adult Female Offspring

**DOI:** 10.3390/ijms25169052

**Published:** 2024-08-21

**Authors:** Asdrubal Aguilera-Méndez, Ian Figueroa-Fierros, Xóchilt Ruiz-Pérez, Daniel Godínez-Hernández, Alfredo Saavedra-Molina, Patricia Rios-Chavez, Santiago Villafaña, Daniel Boone-Villa, Daniel Ortega-Cuellar, Marcia Yvette Gauthereau-Torres, Renato Nieto-Aguilar, Zoraya Palomera-Sanchez

**Affiliations:** 1Instituto de Investigaciones Químico Biológicas, Universidad Michoacana de San Nicolás de Hidalgo, Morelia 58030, Mexico; 1414618f@umich.mx (I.F.-F.); xochitl.ruiz@umich.mx (X.R.-P.); daniel.godinez@umich.mx (D.G.-H.); francisco.saavedra@umich.mx (A.S.-M.); 2Facultad de Biología, Universidad Michoacana de San Nicolás de Hidalgo, Morelia 58030, Mexico; patricia.rios@umich.mx; 3Sección de Estudios de Posgrado e Investigación, Escuela Superior de Medicina, Instituto Politécnico Nacional, Ciudad de Mexico 11340, Mexico; svillafana@ipn.mx; 4Escuela de Medicina, Unidad Norte, Universidad Autónoma de Coahuila, Piedras Negras 26090, Mexico; danielboone@uadec.edu.mx; 5Laboratorio de Nutrición Experimental, Instituto Nacional de Pediatría, Secretaría de Salud, Ciudad de Mexico 04530, Mexico; dortegadan@gmail.com; 6Facultad de Medicina y Ciencias Biológicas, Universidad Michoacana de San Nicolás de Hidalgo, Morelia 58020, Mexico; marcia.gauthereau@umich.mx; 7Facultad de Odontología, Centro Universitario de Estudios de Postgrado e Investigación, Universidad Michoacana de San Nicolás de Hidalgo, Morelia 58330, Mexico; rnieto@umich.mx; 8Facultad de Medicina Veterinaria y Zootecnia, Universidad Michoacana de San Nicolás de Hidalgo, Morelia 58130, Mexico; zoraya.palomera@umich.mx

**Keywords:** biotin, metabolic syndrome, intrauterine growth restriction, fetal programming, undernutrition

## Abstract

Numerous studies indicate that intrauterine growth restriction (IUGR) can predispose individuals to metabolic syndrome (MetS) in adulthood. Several reports have demonstrated that pharmacological concentrations of biotin have therapeutic effects on MetS. The present study investigated the beneficial effects of prenatal biotin supplementation in a rat model of intrauterine caloric restriction to prevent cardiometabolic risk in adult female offspring fed fructose after weaning. Female rats were exposed to a control (C) diet or global caloric restriction (20%) (GCR), with biotin (GCRB) supplementation (2 mg/kg) during pregnancy. Female offspring were exposed to 20% fructose (F) in drinking water for 16 weeks after weaning (C, C/F, GCR/F, and GCRB/F). The study assessed various metabolic parameters including Lee’s index, body weight, feed conversion ratio, caloric intake, glucose tolerance, insulin resistance, lipid profile, hepatic triglycerides, blood pressure, and arterial vasoconstriction. Results showed that GCR and GCRB dams had reduced weights compared to C dams. Offspring of GCRB/F and GCR/F dams had lower body weight and Lee’s index than C/F offspring. Maternal biotin supplementation in the GCRB/F group significantly mitigated the adverse effects of fructose intake, including hypertriglyceridemia, hypercholesterolemia, hepatic steatosis, glucose and insulin resistance, hypertension, and arterial hyperresponsiveness. This study concludes that prenatal biotin supplementation can protect against cardiometabolic risk in adult female offspring exposed to postnatal fructose, highlighting its potential therapeutic benefits.

## 1. Introduction

Metabolic syndrome (MetS) is a cluster of cardiometabolic risk factors that include abdominal obesity, insulin resistance, hypertension, hepatic steatosis, and dyslipidemia. MetS is significantly associated with an increased risk of developing type 2 diabetes and cardiovascular diseases [1]. These chronic non-communicable diseases (NCDs) constitute the leading cause of morbidity and mortality worldwide [2]. The origins of susceptibility to many NCDs in adults, including MetS, could originate from early life through what is known as the “developmental origins of health and disease” (DOHaD) [3,4].

Maternal nutritional status during pregnancy is an important determinant of fetal growth and development, where its deterioration can lead to the development of MetS. Epidemiological and animal studies have consistently demonstrated a negative relationship between the maternal environment and adult metabolic diseases [3,4]. If the fetus is vulnerable to the maternal environment during intrauterine development, then the offspring of parents with MetS are at higher risk of developing this disease, regardless of sex or age [5]. Furthermore, maternal undernutrition is linked to the induction of intrauterine growth restriction (IUGR) and is one of the leading causes of perinatal morbidity, affecting approximately 7% to 15% of pregnancies worldwide [6]. Therefore, low birth weight is related to “fetal programming”, which may determine the onset of future health problems [6,7]. Fetal programming allows the new organism to maintain homeostasis under inadequate conditions, by redistributing nutrients to the most vital organs, resulting in altered organogenesis. While this may be beneficial for short-term survival, it may become detrimental later in life if there is a mismatch between the prenatal and postnatal environments [3,4,7]. In addition, several insult studies in developmental models have shown differences in the response to developmental programming between sexes [8]. Thus, when deprivation is followed by abundance, catch-up growth occurs, predisposing individuals to the development of metabolic diseases [9,10].

Treatment strategies for MetS include weight loss, adoption of a healthy lifestyle, and pharmacological agents, but total recovery remains unreported. In addition to side effects, drug treatments are not always satisfactory for the control of metabolic complications [11]. Therefore, it is essential to explore therapeutic alternatives that are both more effective and user-friendly. Abundant studies have provided evidence of the beneficial effects of vitamins at pharmacological concentrations for the management of MetS [12].

Biotin is a water-soluble vitamin of B-complex that acts as a prosthetic group of carboxylases, which play a key role in a variety of biochemical functions, including the metabolism of carbohydrates, fatty acids, and amino acids [12,13,14,15]. Numerous preclinical and epidemiological studies have shown that biotin at pharmacological concentrations (2 mg–100 mg, 30 to 650 times the daily requirement) has hypoglycemic, hypotriglyceridemic, and antihypertensive effects [12,13,15]. Additionally, no toxic effects of biotin have been reported at these concentrations and its effects are related to modifications at transcriptional, translational, and post-translational levels [12,15]. However, the effect of biotin on fetal programming has not been studied. Therefore, this study aims to investigate the beneficial effects of prenatal biotin supplementation in a rat model of intrauterine caloric restriction to prevent cardiometabolic risk in adult female offspring exposed to a chronic fructose diet postnatally.

## 2. Results

### 2.1. Food Intake and Body Weight in Dams and Body Weight Gain, Caloric Intake, Feed Conversion Ratio, and Lee’s Index in Female Offspring

Dams subjected to global caloric restriction exhibited a significant 13% decrease in body weight during gestation compared to their control counterparts. No significant differences were observed between the GCRB and C groups. The female offspring were fed for 16 weeks according to each group, C, C/F, GCR/F, GCRB/F, starting at 21 days of age. In all groups that consumed fructose, food consumption decreased compared to the control group. The progeny within GCR/F and GCRB/F groups displayed significantly lower weaning weights compared to control groups (C and C/F). While GCR/F and GCRB/F groups had similar final body weights, both were reduced relative to control groups. Interestingly, the C/F group displayed increased body weight and Lee’s index compared to other restriction-diet groups. Weight gain was greater in the fructose-fed groups from undernourished mothers compared to the control group, as was caloric intake. The feed conversion ratio (FCR) was lower in the fructose-fed groups originating from undernourished mothers. Litter size remained unaffected by nutritional restriction, and maternal nursing behavior was consistent across groups (Table 1).

### 2.2. Intraperitoneal Glucose Tolerance and Insulin Resistance Testing

With respect to glucose homeostasis, following a 16-week period post-weaning, notably, the GCR/F and C/F groups exhibited elevated glucose levels at various time points, whereas biotin administration to maternal subjects effectively countered this effect within the GCRB/F offspring group, aligning their glucose levels with the C group (Figure 1a,b). Similarly, biotin improved insulin resistance, by reducing serum glucose levels within the GCRB/F group during specific time intervals (Figure 1b,c).

### 2.3. Serum Lipid Profile Parameters

Regarding lipid profiles, the GCR/F group exhibited dyslipidemia characterized by a noteworthy increase in serum triglyceride, cholesterol, and LDL levels, and a significant decrease in serum HDL levels when compared to the C group (Figure 2). Regarding the C/F group, there was an increase in triglyceride levels compared to the other groups. In the group of offspring from mothers with caloric restriction but supplemented with biotin, the triglyceride, cholesterol, LDL, and HDL values were similar to those of the C group (Figure 2).

### 2.4. Hepatic Triglyceride Quantification

We observed significantly higher hepatic triglyceride levels in the GCR/F group compared to all other groups. The C/F group showed a trend towards higher hepatic triglyceride content compared to the C and GCRB/F groups, although this difference did not reach statistical significance. In contrast, the GCRB/F group exhibited a hepatic triglyceride concentration comparable to that of the C group (Figure 3).

### 2.5. Arterial Blood Pressure Measurement

An increase in systolic and diastolic blood pressure (BP) was observed in the GCR/F group compared to the other groups, and it was slightly higher in the C/F group compared to the C and GCRB/F groups. Interestingly, the GCRB/F group exhibited BP values following a pattern similar to those of the C group (Figure 4).

### 2.6. Measurement of Vascular Reactivity

Offspring fed with fructose from calorie-restricted dams (GCR/F) exhibited concentration-dependent vascular responses to phenylephrine in isolated thoracic aortic rings, both with (Figure 5a) and without endothelium (Figure 5b), resulting in heightened reactivity compared to the other groups. Offspring from dams with unrestricted diets, but fructose-fed (C/F group), displayed slightly increased aortic contractile responses to phenylephrine (PE), although these were lower than in the GCR/F group. Notably, the GCRB/F group, born to calorie-restricted but biotin-supplemented dams, exhibited aortic contractile responses similar to those of the C group (Figure 5a,b).

## 3. Discussion

A growing body of research shows that maternal malnutrition during pregnancy increases the risk of metabolic syndrome in offspring [4,5,6]. The current study represents a pioneering investigation into the therapeutic potential of maternal biotin supplementation in rat models experiencing malnutrition. The results indicate that adult female offspring from malnourished mothers supplemented with biotin experienced a reversal of comorbidities associated with MetS, including insulin resistance, dyslipidemia, hepatic steatosis, hypertension, and arterial hyperresponsiveness. Furthermore, the weight of the biotin-treated mothers was not affected by caloric restriction. Therefore, prenatal biotin supplementation has the potential to serve as a preventive strategy for MetS induced by global caloric restriction during pregnancy.

In this study, dams from the caloric-restricted group (GCR) had lower final body weights than the other groups, consistent with reports that caloric restriction during pregnancy leads to maternal weight loss [16,17,18,19]. The prenatal biotin supplementation maintains maternal weight gain at levels close to control values. Biotin probably compensates for the metabolic imbalance, suggesting that this treatment may have a protective effect against intrauterine malnutrition. Furthermore, all litters reached full gestation and litter size was unaffected.

The extent of fetal growth retardation achieved by maternal diet restriction was akin to that witnessed in other rat models of IUGR [16,17,18]. Changes in body weight gain and Lee’s index were noted at 16 weeks in the C/F group, with all groups exhibiting similar final weights due to catch-up growth. Many animal studies have shown a strong association between suboptimal nutrition during fetal life and postnatal catch-up growth, with lasting adverse effects across a broad phenotypic spectrum, including metabolic syndrome [20,21]. However, the introduction of fructose to promote rapid catch-up growth did not induce obesity in the GCR/F and GCRB/F groups. This divergence may stem from the active behavior and heightened metabolic rate of Wistar rats [22]. Nevertheless, it became apparent that catch-up growth exacerbated the adverse effects of maternal restriction [20,21,23]. Fructose consumption resulted in diminished food intake, consistent with prior findings [14,24]. This decrease is likely attributed to the fact that the extra high-fructose diet increased total caloric intake [14,22], as observed in the fructose-consuming groups, and metabolic programming did not modify this trend. However, a lower feed conversion ratio (FCR) was observed in the fructose-consuming groups compared to the control, suggesting greater feed efficiency, as these groups consumed less food while achieving a higher weight gain per day.

Several studies have shown that chronic fructose consumption has been linked to insulin resistance and metabolic syndrome [24]. Furthermore, progeny exposed to IUGR exhibit heightened insulin resistance, with impaired insulin sensitivity linked to low birth weight [25]. In our study, we observed that in the C/F group, fructose increased in serum glucose concentrations in both the glucose tolerance and insulin resistance curves, according to other studies [14,22]. This effect was more pronounced within the GCR/F group. Conversely, biotin treatment significantly reduced fasting glucose levels and improved glucose tolerance in the GCRB/F group. Numerous studies have shown that pharmacological doses of biotin have hypoglycemic effects, both in patients and murine models [12,15]. This effect is related to a decrease in the expression of gluconeogenic genes in the liver (PEPCK, glucose-6-phosphatase, transcription factors FoxO-1, and HNF-4-alpha), and it has a positive impact on hepatic and pancreatic glucokinase mRNA expression and activity. Furthermore, biotin positively affects pancreatic endocrine function, gene expression, and insulin secretion, ultimately improving insulin sensitivity [12,15]. In the rat liver, biotin has been shown to reduce lipoperoxidation, which may influence the production of reactive oxygen species [14].

Another characteristic component of MetS is dyslipidemia and hepatic steatosis [1,11] observed with chronic fructose consumption [22,24]. This is due to an increase in hepatic lipogenesis and subsequent lipid accumulation, caused by an imbalance between hepatic lipid acquisition and removal [24,26]. In the present study, IUGR rats (GCR/F group) showed a significant increase in serum and hepatic triglyceride concentration, an increase in LDL, and a decrease in HDL levels. The fructose-fed control group exhibited similar behavior but with lower values than the GCR/F group. Notably, female rats demonstrated greater resistance to hepatic steatosis development, probably due to the influence of estrogens on hepatic metabolism [27,28]. However, biotin supplementation in the GCRB/F group decreased hepatic and serum triglyceride concentrations to levels similar to those of the control group. In addition, biotin decreased LDL levels and increased HDL levels. Pharmacological concentrations of biotin have hypotriglyceridemic effects, achieved by reducing the expression of lipogenic genes and abundance of proteins in the liver (SREBP1, acetyl-CoA carboxylase-1, fatty acid synthase, and pyruvate kinase), as well as in adipose tissues (SREBP1, acetyl-CoA carboxylase-1, fatty acid synthase, glucose-6-phosphate dehydrogenase, phosphofructokinase-1, and PPAR-gamma) [12,14,15]. Moreover, biotin supplementation decreases serum free fatty acid concentrations [29], blunting lipogenesis and lipoprotein export. Biotin also activates AMPK kinase, a pivotal factor governing lipid synthesis and oxidation, closely associated with conditions linked to MetS [30]. Moreover, biotin supplementation in mice was observed to heighten glucagon expression and secretion without concurrent alterations in fasting blood glucose levels [31]. This elevation has been associated with acutely enhanced hepatic lipid clearance and suppressed de novo lipogenesis [32].

Multiple preclinical and epidemiological studies provide strong evidence linking adverse intrauterine environments and vascular function alterations to an increased risk of adult cardiovascular disease, including hypertension, in adulthood [10,16,17]. Hypertension is a crucial component of MetS [1,11] and different research has shown that long-term fructose administration in rats results in arterial hypertension [14,24]. We found increased arterial pressure in the C/F group, which was even more pronounced in the GCR/F group compared to the C group. However, the blood pressure levels in the GRCB/F group were similar to those in the C group. Similarly, the contraction in arteries with and without endothelium followed a similar pattern to the blood pressure results. Previous studies by Watanabe (2008) [33] and Aguilera (2018, 2019) [13,14] support the hypotensive and vasorelaxant effects of biotin in genetically hypertensive rats and rats with MetS, respectively. The vasorelaxant impact of biotin is proposed to be endothelium- and nitric oxide-independent, involving guanylate cyclase activation [33] and alteration of Ca^2+^ channels [13]. Studies on human patients and experimental models reveal specific mechanisms related to programmed hypertension in the kidney, such as reduced nephron number, oxidative stress, epigenetic regulation, activation of the renin–angiotensin system (RAS), and sodium transporters. Renal programming is recognized as a key driver of programmed hypertension [34]. Malnutrition is also known to raise hypertension risks through mechanisms such as abnormal vascular function and stimulation of the RAS [35]. As a result, biotin could potentially modulate various systems involved in blood pressure regulation, including the kidney, the renin–angiotensin system, stress-induced stimulation of the hypothalamic–pituitary–adrenal axis, insulin resistance, and endothelial damage [14,35].

Epigenetic modifications are essential in linking a mother’s nutrition with the metabolic health of her offspring. The “Developmental Origins of Health and Disease” theory suggests that non-genetic changes in physical traits result from epigenetic modifications like DNA methylation, histone acetylation, and microRNA expression [36,37]. Biotin’s influence on gene expression could involve a specific signaling pathway, to which histone biotinylation may contribute [12,15]. Therefore, our research group is currently conducting experiments to evaluate possible epigenetic modifications in specific genes related to carbohydrate and lipid metabolism through the activation of the PKG signaling pathway or histone biotinylation.

Differential responses to early-life programming based on sex have been observed in various studies, showing varying effects depending on the timing of famine exposure during gestation. Studies focusing on females revealed metabolic syndrome-related issues under malnutrition conditions [27,28]. Therefore, we decided to focus on studying the response in females. Moreover, considering the predominantly male-centric focus in prior investigations concerning the pharmacological impacts of biotin, there arises a necessity to ascertain the extent to which these effects remain consistent regardless of gender. Within the scope of our inquiry, we discerned analogous outcomes among females vis-à-vis males with regards to the antihypertensive, hypolipidemic, and hypoglycemic attributes of biotin. This observation suggests a negligible influence of female hormones on the metabolic effects of biotin on lipid and carbohydrate metabolism. Nevertheless, acknowledging the pronounced physiological and metabolic distinctions inherent to each gender, it becomes imperative for forthcoming inquiries to undertake comparative analyses of biotin’s effects across sexes. These distinctions are likely to influence varying rates of cardiometabolic risk and susceptibility to disease development in both men and women.

In summary, this exploratory study demonstrates for the first time that prenatal biotin supplementation in a rat model of intrauterine growth restriction exerts a protective effect against cardiometabolic risk in female offspring exposed to postnatal fructose feeding. These risks encompass hyperglycemia, insulin resistance, dyslipidemia, hepatic steatosis, hypertension, and arterial hyperresponsiveness, thereby emphasizing its potential therapeutic efficacy.

## 4. Materials and Methods

### 4.1. Experimental Animals

Rats were handled under the established guidelines for the use and care of laboratory animals of Mexico and approved by the Biosecurity Committee of the Biological Chemistry Research Institute/UMSNH (NOM-062-ZOO-1999). The animals were kept under standard environmental conditions (temperature: 25 ± 2 °C, light/dark cycle of 12 h) and were fed a standard diet (calories provided as a percentage: 58.3 carbohydrates, 13.1 fat, and 28.5 protein) containing biotin at a concentration of 0.00003 g/kg (Labdiet 5012, LandOLakes, Inc., Arden Hills, MN, USA).

### 4.2. Experimental Design

Female and male rats (3 months old) were weight-matched and housed together in a 2:1 ratio for mating for 5 days. If a vaginal plug was detected, the females were single-housed, and the time was recorded as embryonic day 0.5. Thereafter, the dams were randomly divided into three groups (*n* = 4) and housed in standard cages with two rats per cage in each group. They were placed in one of the following diet groups: a control group (C) receiving a standard diet ad libitum; or a global caloric restriction group (GCR), where food intake for the restricted rats (20%) was calculated based on the amount consumed by their respective pair mates in the ad libitum-fed groups. The amount of food given to the rats paired at 20% of energy intake was calculated using the following formula: [(food consumed by the ad libitum-fed pair mate during the previous day/weight of this rat on the previous day) × (0.2) × (current weight of the rat for which the food was estimated)]. The GCR group also received biotin (Sigma-Aldrich, St. Louis, MO, USA) (GCRB) at a dose of 2 mg/kg body weight (IP), which represents approximately 35 times the daily requirement for rats [13,15]. The biotin was dissolved in PBS buffer (pH 7.4) and administered daily throughout the gestational period. The dose of biotin was based on previous studies that showed significant antihyperglycemic, antihyperlipidemic, and antihypertensive effects [12,14,15]. Male rats for mating (average 250 ± 10 g) received the standard diet ad libitum. After birth, the litter size was adjusted to 9 pups on postnatal day (PD) 3 to ensure no nutritional bias. Dams nursed the pups until weaning at PD 21 and were free to feed on the standard diet during lactation. The number of offspring per litter was quantified, and body weights were recorded on day 21 of postnatal life to prevent maternal rejection. They were weaned and divided into 3 groups (*n* = 8 per group), with two offspring per dam based on the type of maternal diet (C, GCR, and GCRB) and housed in standard cages with two rats per cage in each group. Female offspring, randomly selected from each litter, were designated to be used in all subsequent experimental procedures and their estrous cycle was not monitored. Throughout 16 weeks, all groups, except for the control group (C), were provided with a 20% fructose (F) solution in tap water ad libitum to induce catch-up growth. Fructose supplementation was quantified by measuring the daily water consumption of rats to which 20% fructose (*w*/*v*) had been added. By knowing the total volume of water consumed, the amount of fructose consumed per rat in grams, and the corresponding caloric intake in kilocalories (kcal), were calculated. Simultaneously, the rats were fed a standard diet. The offspring were categorized as C, GCR/F, and GCRB/F, while a control group (C/F) comprising offspring from mothers fed the standard diet was also incorporated. After this period, the rats underwent a 12 h fasting period and were subsequently euthanized following established guidelines (Figure 6). Blood samples were collected through cardiac puncture after anesthesia (sodium pentobarbital, 55 mg/kg IP). Liver tissue was excised and preserved at −80 °C for subsequent analysis.

### 4.3. Assessment of Food Intake, Body Weight, Body Weight Gain, Caloric Intake, Feed Conversion Ratio, and Lee’s Index

Weights and food intake of dams were measured three times per week during gestation (21 days), and of pups weekly during the last 2 months of treatment. We calculated the average total food intake over the course of 21 days for dams and the last 8 weeks for pups. To determine food intake in dams and pups, the quantity of feed remaining was subtracted from the quantity of feed provided within a 24 h interval. Body weight gain was calculated by subtracting the body weight of the rat on the last day of the 16-week period from its weight at weaning on postnatal day (PD) 21. Daily body weight gain was determined by dividing the total body weight gain by the number of days from 8 weeks to the last day of the 16-week period. Caloric intake was calculated using the following formula: Caloric Intake (Kcal/g/day) = Daily food intake (g) × total energy food of (kcal/kg)/1000 + kcal equivalent to 20% of fructose consumption based on water intake per rat. The energy provided by the standard diet was 3200 kcal/kg. The feed conversion ratio (FCR) was calculated by dividing the feed intake of the offspring by their average daily body weight gain over the last 8 week. Lee’s index was calculated using the following formula: Lee’s index = cube root of body weight (g)/nose-to-anus length (cm) [38].

### 4.4. Assessment of Intraperitoneal Glucose Tolerance and Insulin Resistance Testing

After a 16-week period of post-weaning treatment, for the glucose test, rats were fasted for 10 h and given a single dose of glucose (2 g/kg IP). Blood samples were obtained by tail nick, and glucose concentration was determined at 0, 30, 60, and 120 min after the glucose injection using a glucometer (Accutrend Plus, ROCHE, Basel, Switzerland). For the insulin test, fed rats were injected with 1 IU/kg body weight of regular human insulin (Eli Lilly, Indianapolis, IN, USA) via IP, and blood glucose levels were checked at 0, 30, 60, and 120 min after the glucose injection by the same method.

### 4.5. Serum Triglyceride, Total Cholesterol, LDL, and HDL Concentration Analysis

Fasting serum triglyceride, total cholesterol, LDL, and HDL concentrations were determined using enzymatic colorimetric kits (Spinreact, Girona, Spain) and measured spectrophotometrically, following the manufacturer’s instructions.

### 4.6. Measurement of Hepatic Triglyceride

Total triglycerides were extracted from 100 mg of frozen liver. The samples were homogenized in 1 mL of solution containing 5% Triton-X100 in PBS buffer using a Polytron (Kinematica AG, Littau, Switzerland), and hepatic triglyceride concentrations were assessed using a colorimetric enzymatic method based on the protocol outlined by Aguilera et al. in 2012 [30]. Briefly, samples were heated in a water bath at 50–60 °C for 2–5 min, then cooled to room temperature, and the process was repeated to solubilize triglycerides. The samples were diluted 1:5 in distilled water and centrifuged for 15 min at maximum speed. A commercial kit, GPO-POD Enzymatic-Colorimetric (Spinreact), was used for the analysis according to the manufacturer’s instructions.

### 4.7. Blood Pressure Measurement

Prior to euthanasia, systolic (SBP) and diastolic (DBP) blood pressure were measured using a tail-cuff plethysmography method (CODA tail-cuff blood pressure system, Kent Scientific Corporation, Torrington, CT, USA) in conscious animals, following a previously described procedure [14]. Each blood pressure reading was obtained by averaging three consistent measurements (with a variation of less than 5 mmHg) per rat. The data represent the average of 4 measurements per week over the last month.

### 4.8. Measurement of Vascular Reactivity in Aortic Rings

The experiments were conducted following the procedures described in previous studies [13,14]. The aortic rings were examined with intact endothelium or with endothelium removed. Briefly, the aortic rings were equilibrated and sensitized with a submaximal concentration of phenylephrine (0.1 μM) for 30 min until reaching a resting tension of 3.0 g. Concentration–response curves to phenylephrine were generated by adding cumulative concentrations ranging from 1 × 10^−9^ to 1 × 10^−5^ M. The contraction was measured using isometric force transducers (Grass FT03, Astro-Med, West Warwick, RI, USA) connected to a data acquisition system MP100 (Biopac Systems, Inc., Goleta, CA, USA).

### 4.9. Statistical Analysis

The results were analyzed using a one-way ANOVA to estimate the impact of maternal restriction diet and post-weaning diet as dependent variables. The therapeutic effect of biotin on offspring (mixture composed of the offspring of the four mothers per group) and fructose group was evaluated using a post hoc Tukey test to determine the differences between the four offspring groups when a statistically significant interaction between birth weight and diet was observed. The data for glucose tolerance tests and lipid measurements were analyzed using repeated measures analysis. Data are presented as mean ± S.E.M., with statistical significance assigned to outcomes displaying *p* < 0.05. The entire analysis was executed utilizing SigmaPlot^®^ 11.0 software.

## 5. Conclusions

In summary, this exploratory study demonstrated, for the first time, that prenatal biotin supplementation in a rat model of intrauterine growth restriction exerts a protective effect on certain alterations related to metabolic syndrome. These alterations include hyperglycemia, insulin resistance, dyslipidemia, hepatic steatosis, hypertension, and arterial hypercontraction in female offspring challenged with postnatal fructose feeding.

## Figures and Tables

**Figure 1 ijms-25-09052-f001:**
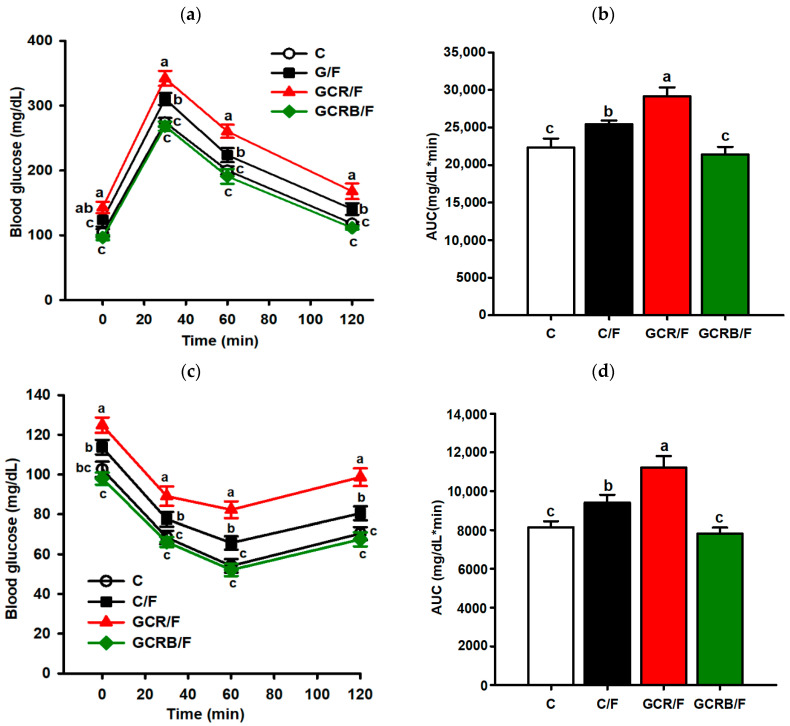
Effect of biotin on blood glucose. (**a**) Intraperitoneal glucose tolerance test. (**b**) Area under the curve (AUC) during intraperitoneal glucose tolerance test. (**c**) Intraperitoneal insulin resistance test. (**d**) Area under the curve during intraperitoneal insulin resistance test. Data (*n* = 8) are presented as means ± SEM. Means without a common letter differ, *p* < 0.05. Statistical significance was determined by repeated measures, followed by the Tukey post hoc test to determine significant differences between treatment groups.

**Figure 2 ijms-25-09052-f002:**
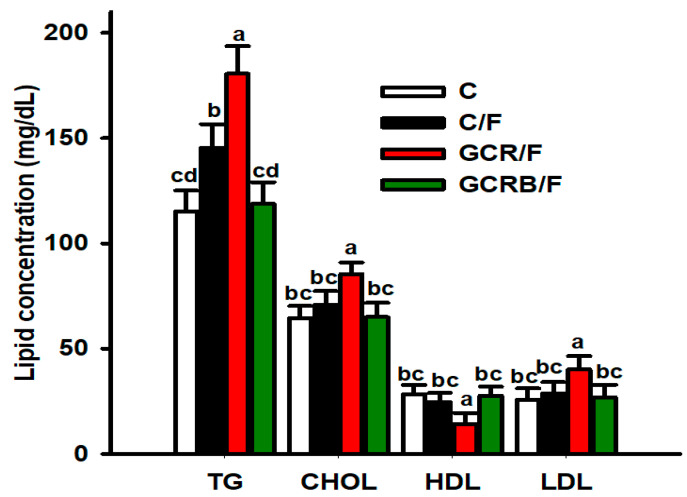
Serum lipid profile. Serum concentrations of triglycerides (TG), cholesterol (CHOL), high-density lipoprotein (HDL), and low-density lipoprotein (LDL). Data (*n* = 8) are presented as means ± SEM. Means without a common letter differ, *p* < 0.05. Statistical significance was determined by repeated measures, followed by the Tukey post hoc test to determine significant differences between treatment groups.

**Figure 3 ijms-25-09052-f003:**
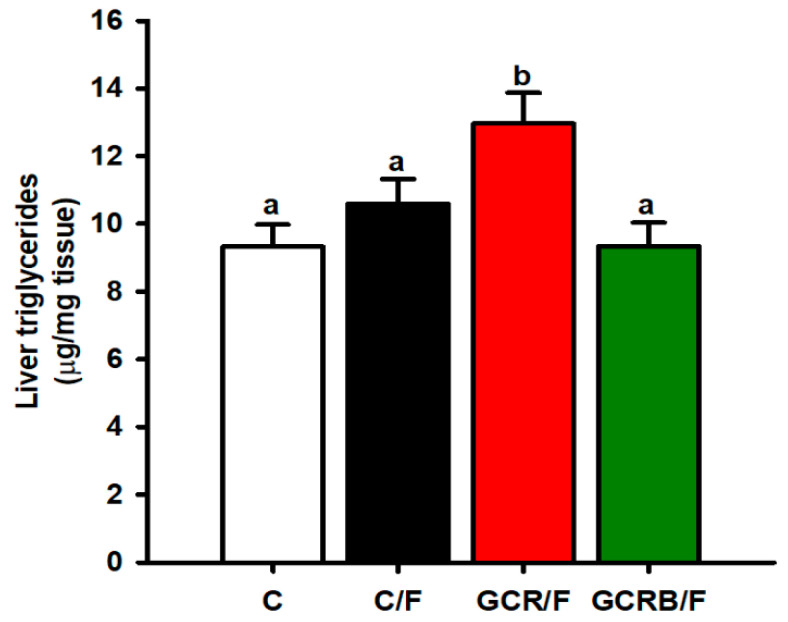
Hepatic lipid concentration. Data (*n* = 8) are presented as means ± SEM. Means without a common letter differ, *p* < 0.05. Statistical significance was determined by one-way ANOVA, followed by the Tukey post hoc test to determine significant differences between treatment groups.

**Figure 4 ijms-25-09052-f004:**
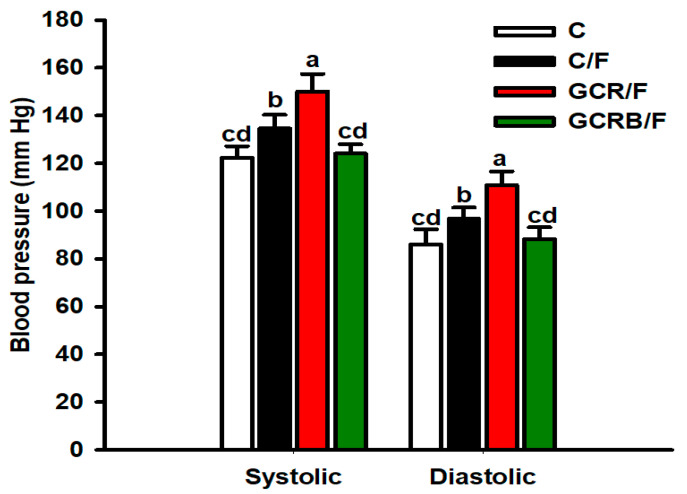
Blood pressure (systolic and diastolic values). Dates (*n* = 8) are mean ± SEM. Means without a common letter differ, *p* < 0.05. Statistical significance was determined using one-way ANOVA, followed by the Tukey post hoc test to determine significant differences between treatment groups.

**Figure 5 ijms-25-09052-f005:**
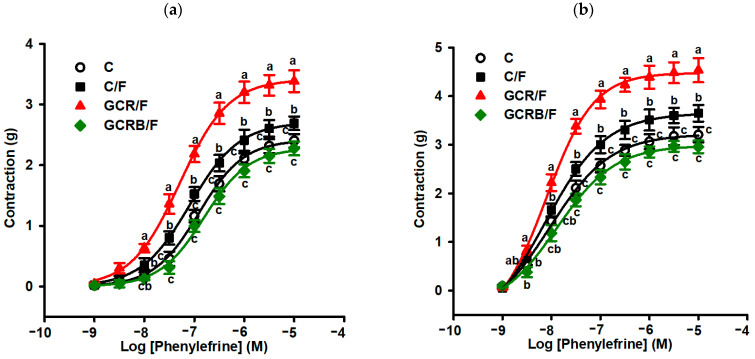
Phenylephrine cumulative concentration-response curves on aortic rings. Curves were generated with endothelium (**a**) and without endothelium (**b**). Means without a common letter differ, *p* < 0.05. Statistical significance was determined using one-way ANOVA and Tukey’s post hoc testing.

**Figure 6 ijms-25-09052-f006:**
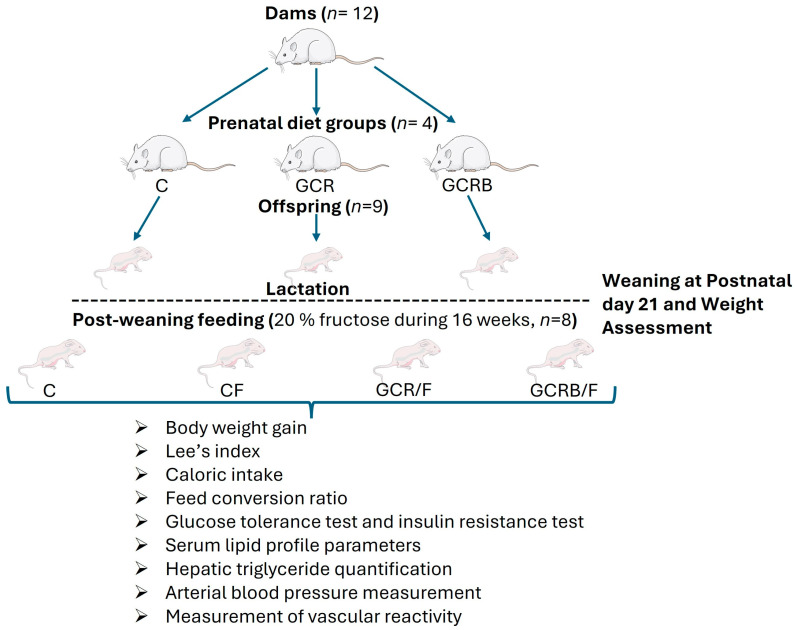
Study design. Prenatal diet groups (dams, *n* = 12), C: control (standard diet); GCR: global caloric restriction group (20%); GCRB: global caloric restriction (20%) + biotin (mg/kg body weight, IP). Lactation (offspring, *n* = 9). Post-weaning feeding (diet standard plus 20% fructose during 16 weeks, *n* = 8). C: control (without fructose); CF: control fructose; GCR/F: global caloric restriction/fructose; GCRB/F: global caloric restriction biotin/fructose. The following online program was used: https://smart.servier.com/ (accessed on 9 August 2024).

**Table 1 ijms-25-09052-t001:** Quantification of various parameters in dams and female offspring.

Variables	Dams Group					
	C		GCR		GCRB	
**Food intake (g/24 h)**	21.5 ± 1.4 ^a^		17.3 ± 0.87 ^b^		17.6 ± 0.92 ^b^	
**Litter size**	11 ± 0.85 ^a^		10 ± 1.08 ^a^		11 ± 0.93 ^a^	
**Males**	5 ± 0.21 ^a^		4 ± 0.38 ^a^		5 ± 0.27 ^a^	
**Females**	6 ± 0.3 ^a^		6 ± 0.56 ^a^		6 ± 0.33 ^a^	
**Initial body weight (g)**	188.8 ± 2.7 ^a^		190.5 ± 3.1 ^a^		187.1 ± 3.5 ^a^	
**Final body weight (g)**	257.6 ± 5.8 ^a^		224.3 ± 7.5 ^b^		254.5 ± 6.1 ^a^	
	**Offspring Group**					
	**C**	**C/F**		**GCR/F**		**GCRB/F**
**Food intake (g/24 h)**	21.8 ± 1.1 ^a^	15.1 ± 0.95 ^b^		16.5 ± 0.81 ^b^		15.8 ± 0.85 ^b^
**Caloric intake (kcal/g/day)**	69.76 ± 2.5 ^a^	84.24 ± 3.7 ^b^		83.55 ± 3.3 ^b^		81.94 ± 3.1 ^b^
**Feed conversion ratio (g/g)**	5.70 ± 0.8 ^a^	3.75 ± 0.6 ^b^		4.18 ± 0.7 ^bc^		4.04 ± 0.9 ^bc^
**BW at weaning (g)**	53.5 ± 2.8 ^a^	52.0 ± 3.1 ^a^		41.8 ± 3.2 ^b^		45.2 ± 2.9 ^b^
**Final body weight (g)**	283.2 ± 7.5 ^a^	293.5 ± 9.9 ^b^		278.6 ± 9.4 ^a^		279.8 ± 8.2 ^a^
**Body weight gain (g/day)**	3.82 ± 0.1 ^a^	4.02 ± 0.1 ^b^		3.94 ± 0.3 ^b^		3.91 ± 0.2 ^b^
**Lee’s index**	0.30 ± 0.011 ^a^	0.31 ± 0.08 ^b^		0.30 ± 0.09 ^a^		0.30 ± 0.07 ^a^

Food intake: For dams, the average total intake over the course of gestation (21 days) was measured, while for offspring, the average intake over the last 8 weeks was recorded. Values are presented as means ± SEM. Dams: *n* = 4 and offspring: *n* = 8. Means in a row with superscripts without a common letter differ, *p* < 0.05. For dams, a one-way ANOVA was performed, and the Tukey post hoc test was used to determine significant differences between treatment groups in the offspring groups. C = control; GCR= global caloric restriction; GCRB = global caloric restriction biotin; C/F = control/fructose; GCR/F = global caloric restriction/fructose; GCRB/F = global caloric restriction biotin/fructose.

## Data Availability

The data presented in this study are available on request from the corresponding author.

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
