# Peer review of "The Beneficial Effects of Prenatal Biotin Supplementation in a Rat Model of Intrauterine Caloric Restriction to Prevent Cardiometabolic Risk in Adult Female Offspring"

_ijms, 2024, doi:10.3390/ijms25169052_

Round 1

Reviewer 1 Report

Comments and Suggestions for Authors

The present study is interesting and relevant, demonstrating a link between maternal malnutrition during pregnancy and the increased risk of metabolic syndrome-related diseases in offspring. It also highlights the importance of prenatal biotin supplementation in mitigating this relationship. To improve this article, I have the following suggestions:

Specify in both Table 1 and the legend of each figure what the lowercase letters a, b, c, and their combinations represent.

Considering the complexity of the experimental model, I suggest performing a diagram of the experimental model to facilitate quick understanding.

The discussions are extensive and difficult to follow in relation to the results. To address this issue, I suggest performing a heat map representing the variation of selected parameters shown as the fold change relative to control.

Additionally, considering that the discussions are sometimes tangential to the obtained results, a conclusion that reflects the main findings is necessary.

Comments on the Quality of English Language

Minor editing of English language required.

Author Response

Response to Reviewer 1 Comments

Comments 1:

Specify in both Table 1 and the legend of each figure what the lowercase letters a, b, c, and their combinations represent.

Response 1: In both the table and the figures, the meanings of the letters are indicated. In the new version, they are highlighted in red to ensure that the reviewer can easily identify them.

Comments 2:

Considering the complexity of the experimental model, I suggest performing a diagram of the experimental model to facilitate quick understanding.

Response 2:

A diagram of the experimental model is included in Section 4.2, 'Experimental Design,' at the end of the text.

Comments 3:

The discussions are extensive and difficult to follow in relation to the results. To address this issue, I suggest performing a heat map representing the variation of selected parameters shown as the fold change relative to control.

Response 3:

We have summarized the discussion and included the changes in the quantified parameters, presented as the time points of change relative to the control.

Comments 4:

Additionally, considering that the discussions are sometimes tangential to the obtained results, a conclusion that reflects the main findings is necessary.

Response 4:

We have included a conclusion at the end of the discussion

Additional clarifications

We conduct a thorough proofreading of the writing and spelling in the document.

Reviewer 2 Report

Comments and Suggestions for Authors

General Comment.

This is an interesting study and well conducted. The authors have consistently demonstrated that exogenous administration of biotin to post-weaning female rat offspring of mothers under gestational caloric restriction can improve biological markers of metabolic syndrome such as altered carbohydrate metabolism, hepatic dyslipidemia and steatosis, increased blood pressure, and increased arterial response to challenge with phenylephrine. The introduction explains the relevance of addressing the hypothesis, the methods used are adequate and the results clearly support the conclusions of the study. However, the discussion seems to be a bit longer than it needs to be. The bibliography is correctly formatted and cited.

In addition, some relevant issues need to be clarified.

Specific Points

1.       Title and general MetS definition. Metabolic syndrome is not a disease. It is a syndrome: a set of biological markers of pathophysiological conditions described in humans that may be linked to different pathologies but not to any etiological factor. The "metabolic syndrome" is related to cardiovascular risk and mortality in "humans". It should not be applied in experimental animals, although some animal models may resemble metabolic syndrome. Such animal models can be attributed as "metabolic syndrome models." Please mention MetS accordingly in the title and throughout the entire text of the article, and avoid considering MetS as a "chronic disease" or "metabolic syndrome diseases". Accordingly, the title should be changed. Title suggestion: "The beneficial effects of postnatal biotin supplementation in a rat model of intrauterine caloric restriction to prevent metabolic risks in adult female offspring", or any other variation that authors may prefer. Please, include the term “female” in the title, as the experiment was restricted to female offspring and males were ignored.

2.       Results: Table 1 data.

a.       How was the food intake measured both in mothers and pups? Describe in methods.

b.      Once the rat´s body weight may change, the food intake must weighted as g/100gBW (grams/100 grams body weight).

c.       Since fructose supplementation results in an increase in caloric intake, the caloric intake data for the offspring groups should be displayed.

d.      Since caloric intake would be different and final weight of the animals also, the Food conversion index calculation may help to the interpretation of the results.

e.      Why have the authors not shown the comparison of food intake and weight changes with males? The metabolic responses of males and females to gestational food restriction may be very different.

f.        Litter size. Was the whole number of pups (males & females) registered for each mother?? Please, show the results for each group, as that may help to understand the effects of biotin in weight changes of the mothers. A number is shown in the table 1. Is that the mean of pups per mother? Please, show also the SEM. Have the authors recorded the weight of pups at birth?? Males and females?? Litter weight at weaning was still lower in the GCR groups.

3.       Results. Glucose and Insulin tolerance test.

a.       Insulin tolerance is preferred over Insulin resistance, along the text.

b.      The area under the curve (AUC) of the blood glucose levels profile in response to glucose and insulin tolerance test would be very demonstrative of the biotin effects and very easy for the reader to understand. It may be displayed in the side of time curves.

4.       Discussion. As a whole, the discussion seems to be too long. Please, make an effort to shorten it as much as possible.

a.       Lines 193-199. It seems like a weak explanation. How can biotin compensate for reduced caloric intake????. GCR mothers are eating about 20% less food in grams. However, they are producing the same number of puppies. Is the food restriction compensated by the reduction in litter weight at birth (females & males)? These mothers reached a significantly lower weight at the end of the experiment (about 13%). Is the body composition of mothers changing? Are they losing more fat? Consider: the standard composition of the diet is low in fat and high in protein (see comment on methods), so food restriction is mainly reducing fat calories, which can be easily compensated by maternal fat stores. Do the authors agree? For this explanation, the weight of the offspring at birth appears to be necessary. On the other hand, does biotin contribute in any way to mobilizing lipids from mothers' deposits? Comment.

b.      GCR/F pups are eating about 25% fewer grams of food, however, since they are supplemented with fructose, they are likely eating excess calories from fructose (carbohydrates). Fructose itself has deleterious metabolic effects. How much fructose did the animals ingest? Can the authors calculate the actual caloric intake of offspring groups.

c.       Lines 262-266. Biotin-induced glucagon elevation may be involved in increasing circulating glucose levels and IR. Glucagon elevation is a pathogenic factor in human diabetes. Please comment on this further.

d.      What about males? The effects of gestational food restriction are known to be very different in males and females.

5.       Figures.

a.       The quality of the figures should be improved, for example, by using colors for the bars, and selecting other symbols-lines easier to identify in the time curves.

6.       Methods:

a.       How were the rats housed? In standard or metabolic cages? Were the animals housed individually or in groups? How many per cage? Please, specify.

b.      How was the food intake measured in dams and pups?? Please, describe elsewhere.

c.       Composition of the diet. The standard diet fed to the rats was low in fat (13.1%) and high in protein (28.5%), as a percentage of the total calories provided by the diet, compared to the recommended ratios of 25-30% (fat) and 15-20% (prot). The addition of fructose ad libitum turned the diet into even more low-fat and now high-carbohydrate diet. This should be discussed to interpret the results.

d.      Fructose supplementation. Was it measured in any way? Explain how elsewhere.

e.      Method for triglyceride quantification in the liver? Explain it, please.

f.        In which samples was measured the lipid profile? Describe elsewhere.

g.       The two-way ANOVA allows simultaneously evaluating the effect of 2 independent conditions on the dependent variable, considering the effect of one of the variables, the effect of the other or the interaction between them. This requires having at least four experimental groups that combine both conditions. In the case of mothers, there are 2 experimental conditions (caloric restriction and biotin supplementation) but one group (C+biotin) is missing, so two-way ANOVA is not applicable. With regard to offspring, in fact, there are 3 experimental conditions: intrauterine caloric restriction, supply of biotin to mothers, and postnatal fructose supplementation. That can allow even a 3-way ANOVA to be used, however, several groups are also missing, which also makes not possible to use the two-way ANOVA. The calculation of the two-factor ANOVA usually includes the calculation of the more conventional one-factor ANOVA. The one-way ANOVA is perfectly applicable to the experimental groups in the study and it is free from the requirements for using the two-way ANOVA. Since the simultaneous effects of the experimental conditions, nor their interactions, cannot be tested, and the requirements for performing the two-way ANOVA do not seem to be possible to match with the experimental design of the study, the one-way ANOVA, followed by Tukey's post-hoc test for multiple comparisons, seems to be sufficient and perfectly adequate to evaluate the results of the study. This referee surmises that the results shown were those of the one-factor ANOVA. If that's the case, simply substitute "two-way" for "one-way" elsewhere in the text.

h.       The use of the LSD post-hoc test is mentioned in the method, but is not indicated in any of the legends in the figure. Please clarify this point or delete the mention of the LSD test if it is not really used.

Minor points.

Line 41. Consider substitute "arterial hypercontraction" by other more suitable term. It may be "arterial constriction", "arterial hyperresponsiveness to…", ....

Line 42. “MetS-related diseases”. Consider substitute by "metabolic risks", "cardiovascular dysfunction", “MetS biological markers”,…..

Line 57. MetS is not a chronic disease. Please, suppress “chronic disease” in this statement.

Author Response

 Response to Reviewer 2 Comments

Specific Points

Comments 1:

Title and general MetS definition. Metabolic syndrome is not a disease. It is a syndrome: a set of biological markers of pathophysiological conditions described in humans that may be linked to different pathologies but not to any etiological factor. The "metabolic syndrome" is related to cardiovascular risk and mortality in "humans". It should not be applied in experimental animals, although some animal models may resemble metabolic syndrome. Such animal models can be attributed as "metabolic syndrome models." Please mention MetS accordingly in the title and throughout the entire text of the article, and avoid considering MetS as a "chronic disease" or "metabolic syndrome diseases". Accordingly, the title should be changed. Title suggestion: "The beneficial effects of postnatal biotin supplementation in a rat model of intrauterine caloric restriction to prevent metabolic risks in adult female offspring", or any other variation that authors may prefer. Please, include the term “female” in the title, as the experiment was restricted to female offspring and males were ignored.

Response 1:

We appreciate the comment and have revised the title according to the reviewer's suggestion, which is now as follows: “The beneficial effects of prenatal biotin supplementation in a rat model of intrauterine caloric restriction to prevent cardiometabolic risk in adult female offspring”. Additionally, we have revised the text to refer to metabolic syndrome itself rather than its associated diseases, emphasizing its characterization as a metabolic risk.

Comments 2:

2. Results: Table 1 data.

a. How was the food intake measured both in mothers and pups? Describe in methods.

Response 2:

The methodology for quantifying the food consumption of both mothers and offspring is detailed in the methods section (highlighted in red): “To determine food intake in dams and pups, the quantity of feed remaining was subtracted from the quantity of feed provided within a 24-hour interval”. This methodology involved weighing the difference between the amount of feed added and the amount collected in the cages. It is important to note that a pilot test of the 3-day average consumption was conducted prior to this study to establish an accurate amount of feed. Generally, the amount of feed found in the boxes was minimal, resulting in negligible variations in the measurements.

Comments 3:

b. Once the rat´s body weight may change, the food intake must weighted as g/100gBW (grams/100 grams body weight).

Response 3:

We appreciate the reviewer's observation. Some authors present food consumption in various formats, while we, like others, present it in terms of daily consumption. Our aim was to calculate the feed intake as an average over the last 8 weeks for the offspring and 21 days for the dams to observe more substantial changes and to compare it with the quantified weight gain, which is presented in the table for the offspring. Therefore, we believe it is unnecessary to refer to 100 g of body weight, as we have provided body weight data at the beginning and end of the experiment. Additionally, presenting the value of feed consumption per day is more practical for calculating the feed conversion ratio (FCR). As requested by the reviewer, this information has been included in the table for the progeny

Comments 4:

c. Since fructose supplementation results in an increase in caloric intake, the caloric intake data for the offspring groups should be displayed.

Response 4:

We calculated the caloric intake and included it in the table, along with a description of its calculation in the methodology. Additionally, we revised the title in the methodology section.

Comments 5:

d. Since caloric intake would be different and final weight of the animals also, the Food conversion index calculation may help to the interpretation of the results.

Response 5:

We have included the feed conversion ratio (FCR) for the offsprings in the table and described its calculation in the methodology. Additionally, we have referenced it in the discussion.

Comments 6:

e. Why have the authors not shown the comparison of food intake and weight changes with males? The metabolic responses of males and females to gestational food restriction may be very different.

Response 6:

As the reviewer notes, the metabolic response to dietary restriction during pregnancy differs significantly between males and females, as demonstrated in numerous studies. While it is indeed intriguing to analyze the responses arising from sexual dimorphism, this study focused specifically on females. As discussed: “Differential responses to early-life programming based on sex have been observed in various studies, showing varying effects depending on the timing of famine exposure during gestation. Studies focusing on females revealed metabolic syndrome-related issues under malnutrition conditions [28,29]. Therefore, we decided to focus on studying the response in females. Moreover, considering the predominantly male-centric focus in prior investigations concerning the pharmacological impacts of biotin, there arises a necessity to ascertain the extent to which these effects remain consistent regardless of gender. Within the scope of our inquiry, we have discerned analogous outcomes among females vis-à-vis males with regards to the antihypertensive, hypolipidemic, and hypoglycemic attributes of biotin. This observation suggests a negligible influence of female hormones on the metabolic effects of biotin on lipid and carbohydrate metabolism. Nevertheless, acknowledging the pronounced physiological and metabolic distinctions inherent to each gender, it becomes imperative for forthcoming inquiries to undertake comparative analyses of biotin's effects across sexes. These distinctions are likely to influence varying rates of cardiometabolic risk and susceptibility to disease development in both men and women”… In addition, we chose to concentrate on females due to the experimental complexities associated with analyzing both sexes. The number of individuals required to maintain two rats per box and effectively quantify food intake and other parameters results in a substantial demand for space, which poses a limitation in the biotherium for adequately housing the animals. This limitation affected the development of the present study.

Comments 7:

f.  Litter size. Was the whole number of pups (males & females) registered for each mother?? Please, show the results for each group, as that may help to understand the effects of biotin in weight changes of the mothers. A number is shown in the table 1. Is that the mean of pups per mother? Please, show also the SEM. Have the authors recorded the weight of pups at birth?? Males and females?? Litter weight at weaning was still lower in the GCR groups.

Response 7:

The number of male and female offspring per group has been added to the table. The standard error of the mean (SEM) is expressed as the average number of offspring per group of mothers. As indicated in the methodology, offspring body weights were recorded on postnatal day 21 to minimize the risk of maternal rejection; they were not weighed at birth. As discussed below, we do not consider it necessary to weigh the offsprings at birth.

Comments 8:

3. Results. Glucose and Insulin tolerance test.

a. Insulin tolerance is preferred over Insulin resistance, along the text.

Response 8:

We have changed the title and text to “Glucose tolerance and insulin resistance testing”.

Comments 9:

b. The area under the curve (AUC) of the blood glucose levels profile in response to glucose and insulin tolerance test would be very demonstrative of the biotin effects and very easy for the reader to understand. It may be displayed in the side of time curves.

Response 9:

The graphs corresponding to the calculation of the area under the curve (AUC) for the glucose tolerance and insulin resistance tests have been added. And a description was added to the figure legend."

Comments 10:

4. Discussion. As a whole, the discussion seems to be too long. Please, make an effort to shorten it as much as possible.

Response 10:

We have attempted to summarize the discussion by being more concise with the main ideas, without compromising the quality of the content.

Comments 11:

a.  Lines 193-199. It seems like a weak explanation. How can biotin compensate for reduced caloric intake????. GCR mothers are eating about 20% less food in grams. However, they are producing the same number of puppies. Is the food restriction compensated by the reduction in litter weight at birth (females & males)? These mothers reached a significantly lower weight at the end of the experiment (about 13%). Is the body composition of mothers changing? Are they losing more fat? Consider: the standard composition of the diet is low in fat and high in protein (see comment on methods), so food restriction is mainly reducing fat calories, which can be easily compensated by maternal fat stores. Do the authors agree? For this explanation, the weight of the offspring at birth appears to be necessary. On the other hand, does biotin contribute in any way to mobilizing lipids from mothers' deposits? Comment.

Response 11:

With the data obtained in this research, it is challenging to adequately explain how biotin can compensate for the reduction in caloric intake in undernourished mothers. More experiments related to the evaluation of signaling pathways, hormones, and gene expression would be necessary for this purpose. However, we believe that this work opens the door for further investigation and the potential for explanations. We can speculate that one possibility is that biotin may directly increase the activity of carboxylases such as ACC1, which is involved in lipid synthesis, and ACC2, which is associated with lipid oxidation, as well as pyruvate carboxylase, which is related to gluconeogenesis. This could enhance and increase the activity of these metabolic pathways, potentially compensating for lipid, glucose, and energy content, which is essential for the metabolic activities of the mother and the development of the fetus. Consequently, this would eliminate the need for mobilizing triglyceride content from adipose tissue to compensate for reduced caloric intake, thus preventing a decrease in the body weight of biotin-treated undernourished mothers.

Another possibility, as reported in several studies, is that biotin activates the AMPK signaling pathway, which would also enhance the production of necessary energy through the oxidation of fatty acids produced endogenously by an increase in ACC1 activity and the oxidation of glucose produced by gluconeogenesis due to elevated pyruvate carboxylase activity. Additionally, as also reported, biotin increases both insulin and glucagon secretion, which would initially facilitate the uptake of dietary glucose by insulin-dependent cells (adipose and muscle tissue), while glucagon would promote its oxidation, along with that of dietary-derived fatty acids. This would contribute to the lack of mobilization of triglyceride content from adipose tissue, and therefore, a significant reduction in the weight of undernourished mothers would not be observed.

This may explain why biotin-treated undernourished mothers do not experience a reduction in body weight; however, the weight of their offspring does show a decrease compared to the control group, as indicated in the table 1 when they were weighed after weaning, even though the number of offspring at birth was standardized to nine per mother. This did not compensate for the low birth weight. This suggests that while the mother maintained an adequate energy balance to prevent weight loss, the quantity of nutrients supplied to the fetus was insufficient for them to achieve normal birth weight. In the case of undernourished mothers without biotin treatment, they did experience a reduction in weight, and their offspring also had lower weights. In this regard, it has been reported that maternal undernutrition in murine models can lead to reductions in both birth weight and the number of offspring per litter; however, this will depend on the percentage of caloric intake reduction in mothers and their genetic condition, so it is not a rule that intrauterine undernutrition always produces a reduction in litter size. Sometimes it can result in one effect or the other, or both. In this case, only a reduction in weight was observed without a decrease in the number of individuals per litter, which could also be attributed to the percentage of caloric intake in this study being 20%, which is below the 50% typically used in these undernutrition models.

Regarding the necessity of weighing the offspring at birth, we consider it unnecessary, because even after weaning, differences in weights between offspring from biotin-treated and untreated undernourished mothers remain evident compared to the control group, which was not undernourished. This demonstrates that the differences in birth weights are still preserved and observable, even when the number of individuals per litter was reduced to attempt to homogenize the amount of food during lactation.

Comments 12:

b. GCR/F pups are eating about 25% fewer grams of food, however, since they are supplemented with fructose, they are likely eating excess calories from fructose (carbohydrates). Fructose itself has deleterious metabolic effects. How much fructose did the animals ingest? Can the authors calculate the actual caloric intake of offspring groups.

Response 12:

As noted by the reviewer, fructose has detrimental metabolic effects, which we utilized to model the high consumption experienced by malnourished children born to mothers who also suffered from malnutrition. These children typically consume an excess of calories from foods such as juices, sodas, pastries, etc., which are often supplemented with high-fructose corn syrup during their childhood and youth.

The amount of fructose is related to the daily water intake, which contained 20% fructose (weight/volume). Therefore, the amount of water consumed was measured every third day, yielding an average of approximately 38±1.7 mL. From this, the 20% fructose content was calculated, providing the amount in grams consumed by the rats. Knowing that 1 gram of fructose provides 4 kcal, we determined the caloric contribution from fructose based on the amount of water consumed. Subsequently, the daily caloric intake was calculated as follows: Caloric Intake (Kcal/g/day) = Daily food intake (gm) * total energy food of (kcal/kg)/1000 + kcal equivalent to 20% of fructose consumption based on water intake per rat. The energy provided by standard diet was 3,200 kcal/kg. This description was succinctly added to the methodology.

Comments 13:

c.  Lines 262-266. Biotin-induced glucagon elevation may be involved in increasing circulating glucose levels and IR. Glucagon elevation is a pathogenic factor in human diabetes. Please comment on this further.

Response 13:

The study that describes this effect found that dietary biotin supplementation increased glucagon mRNA levels, glucagon release, and fasting plasma glucagon concentration. However, the increase in glucagon levels did not alter fasting blood glucose levels, glycogen degradation, or the expression of the gluconeogenesis rate-limiting enzyme phosphoenolpyruvate carboxykinase. Additionally, previous findings regarding the effects of biotin have reported glucose-stimulated insulin secretion, improved glucose tolerance, and hypotriglyceridemic effects. Therefore, the potentially deleterious effect associated with the increase in glucagon may not be harmful, as several studies have demonstrated that biotin has a beneficial effect on insulin resistance and diabetes.

Comments 14:

d. What about males? The effects of gestational food restriction are known to be very different in males and females.

Response 14:

As mentioned above in response to the reviewer's comment regarding the 2.Results: Table 1 data, e. Why have the authors not shown the comparison of food intake and weight changes with males? The metabolic responses of males and females to gestational food restriction may be very different. We recognize that the metabolic responses of males and females to gestational food restriction may differ significantly, and we believe this issue was adequately addressed in the discussion. We also acknowledged the limitations of our study, specifically the constraints on the number of animals due to the limited space available in the biotherium during our research, which prevented us from analyzing both males and females simultaneously.

Comments 15:

5. Figures.

a. The quality of the figures should be improved, for example, by using colors for the bars, and selecting other symbols-lines easier to identify in the time curves.

Response 15:

The quality of the figures has been improved, as suggested by the reviewer, by using different colors and symbols for easier identification of the curves.

Comments 16:

6. Methods:

a. How were the rats housed? In standard or metabolic cages? Were the animals housed individually or in groups? How many per cage? Please, specify.

Response 16:

The rats were housed in standard cages, with two rats per cage in each group, as specified in the methodology.

Comments 17:

b. How was the food intake measured in dams and pups?? Please, describe elsewhere.

Response 17: As requested by the reviewer, this description has been added to the methodology in Section 4.3: To determine feed intake in dams and pups, the quantity of feed remaining was subtracted from the quantity of feed provided within a 24-hour interval”.

Comments 18:

c. Composition of the diet. The standard diet fed to the rats was low in fat (13.1%) and high in protein (28.5%), as a percentage of the total calories provided by the diet, compared to the recommended ratios of 25-30% (fat) and 15-20% (prot). The addition of fructose ad libitum turned the diet into even more low-fat and now high-carbohydrate diet. This should be discussed to interpret the results.

Response 18:

The standard diet used in our study conforms to the parameters of most standard diets, such as AIN-93M or AIN-93G, which generally contain 10% of energy from lipids, 60.6% from carbohydrates, and 18.6% from protein. In our case, the diet had a higher protein content due to its formulation for pregnancy. As noted by the reviewer, the addition of fructose increased the carbohydrate intake, which was intended to simulate a catch up growth pattern in the offspring. This pattern is associated with an increased risk of metabolic syndrome in adulthood, as highlighted in preclinical and clinical studies. We address this in the Discussion section: " Many animal studies have shown a strong association between suboptimal nutrition during fetal life and postnatal catch-up growth, with lasting adverse effects across a broad phenotypic spectrum, including metabolic syndrome [22,24]. However, the introduction of fructose to promote rapid catch-up growth did not induce obesity in the GCR/F and GCRB/F groups. This divergence may stem from the active behavior and heightened metabolic rate of Wistar rats [21]. Nevertheless, it became apparent that catch-up growth exacerbated the adverse effects of maternal restriction [22-24]. Fructose consumption resulted in diminished food intake, consistent with prior findings [14,25]. This decrease is likely attributed to the fact that the extra high-fructose diet increased total caloric intake [14,21] and metabolic programming did not modify this trend". Given that the reviewer requested a more concise discussion, we believe that the current level of detail is sufficient, and that further elaboration is unnecessary

Comments 19:

d. Fructose supplementation. Was it measured in any way? Explain how elsewhere.

Response 19:

Fructose supplementation was quantified by measuring the daily water consumption of rats to which 20% fructose (w/v) had been added. By knowing the total volume of water consumed, the amount of fructose consumed per rat in grams, and the corresponding caloric intake in kilocalories (kcal) were calculated. This is also indicated in the methodology.

Comments 20:

e. Method for triglyceride quantification in the liver? Explain it, please.

Response 20:

Total triglycerides were extracted from 100 mg of frozen liver. The samples were homogenized in 1 mL of solution containing 5% Triton-X100 in PBS buffer using a Polytron (Kinematica AG, Littau, Switzerland). Samples were subjected to water bath warming at 50–60_C for 2–5 min and cooled down slowly at room temperature. The heating/cooling procedure was repeated to solubilize the triglycerides. Samples were diluted 1:5 in distilled water and centrifuged for 15 min at top speed in a microcentrifuge. A commercial kit, GPO-POD Enzymatic-Colorimetric (Spinreact), was used for the analysis according to the manufacturer's instructions. A brief description has been added to the Methodology section.

Comments 21:

f. In which samples was measured the lipid profile? Describe elsewhere.

Response 21:

The methodology in Section 4.5 describes the measurement of triglycerides, total cholesterol, and LDL and HDL lipoproteins in serum. Section 4.6 indicates that total triglycerides were measured in liver tissue

Comments 22:

g. The two-way ANOVA allows simultaneously evaluating the effect of 2 independent conditions on the dependent variable, considering the effect of one of the variables, the effect of the other or the interaction between them. This requires having at least four experimental groups that combine both conditions. In the case of mothers, there are 2 experimental conditions (caloric restriction and biotin supplementation) but one group (C+biotin) is missing, so two-way ANOVA is not applicable. With regard to offspring, in fact, there are 3 experimental conditions: intrauterine caloric restriction, supply of biotin to mothers, and postnatal fructose supplementation. That can allow even a 3-way ANOVA to be used, however, several groups are also missing, which also makes not possible to use the two-way ANOVA. The calculation of the two-factor ANOVA usually includes the calculation of the more conventional one-factor ANOVA. The one-way ANOVA is perfectly applicable to the experimental groups in the study and it is free from the requirements for using the two-way ANOVA. Since the simultaneous effects of the experimental conditions, nor their interactions, cannot be tested, and the requirements for performing the two-way ANOVA do not seem to be possible to match with the experimental design of the study, the one-way ANOVA, followed by Tukey's post-hoc test for multiple comparisons, seems to be sufficient and perfectly adequate to evaluate the results of the study. This referee surmises that the results shown were those of the one-factor ANOVA. If that's the case, simply substitute "two-way" for "one-way" elsewhere in the text.

Response 22:

We appreciate the reviewer’s comment and clear explanation; we agree and have replaced 'two-way' with 'one-way' throughout the text.

Comments 23:

h. The use of the LSD post-hoc test is mentioned in the method, but is not indicated in any of the legends in the figure. Please clarify this point or delete the mention of the LSD test if it is not really used.

Response 23:

We appreciate the reviewer’s comment and have removed the reference to the LSD test.

Additional clarifications

We conduct a thorough proofreading of the writing and spelling in the document.

Minor points

Line 41. Consider substitute "arterial hypercontraction" by other more suitable term. It may be "arterial constriction", "arterial hyperresponsiveness to…", ....

Response

We have changed the term to 'arterial hyperresponsiveness.

Line 42. “MetS-related diseases”. Consider substitute by "metabolic risks", "cardiovascular dysfunction", “MetS biological markers”,…..

Response

We have changed the term to metabolic risk.

Line 57. MetS is not a chronic disease. Please, suppress “chronic disease” in this statement.

Response

We have suppress “chronic disease” in this statement.

Round 2

Reviewer 1 Report

Comments and Suggestions for Authors

The authors improved the manuscript, achieving the necessary quality for publication.